# Evidence for endogenous hydrogen peroxide production by *E. coli* fatty acyl-CoA dehydrogenase

**Chaiyos Sirithanakorn**[1,2]*, **James A. Imlay**[2]

1 Division of Molecular and Cellular Medicine, King Mongkut's Institute of Technology Ladkrabang, Faculty of Medicine, Bangkok, Thailand, 2 Department of Microbiology, University of Illinois, Urbana, Illinois, United States of America

* Chaiyos.si@kmitl.ac.th

**Data Availability Statement:** All relevant data are within the manuscript and its Supporting information files.

**Funding:** This study was financially supported by King Mongkut's Institute of Technology

## Abstract

Aerobic organisms continuously generate internal superoxide and hydrogen peroxide, which can damage enzymes and impair growth. To avoid this problem cells maintain high levels of superoxide dismutases, catalases, and peroxidases. Surprisingly, we do not know the primary sources of these reactive oxygen species (ROS) in living cells. However, in vitro studies have shown that flavoenzymes can inadvertently transfer electrons to oxygen. Therefore, it seems plausible that substantial ROS may be generated when large metabolic fluxes flow through flavoproteins. Such a situation may arise during the catabolism of fatty acids. Acyl-CoA dehydrogenase (FadE) is a flavoprotein involved in each turn of the beta-oxidation cycle. In the present study the catabolism of dodecanoic acid specifically impaired the growth of strains that lack enzymes to scavenge hydrogen peroxide. The defect was absent from *fadE* mutants. Direct measurements confirmed that the beta-oxidation pathway amplified the rate of intracellular hydrogen peroxide formation. Scavenging-proficient cells did not display the FadE-dependent growth defect. Those cells also did not induce the peroxide stress response during dodecanoate catabolism, indicating that the basal defenses are sufficient to cope with moderately elevated peroxide formation. In vitro work still is needed to test whether the ROS evolve specifically from the FadE flavin site and to determine whether superoxide as well as peroxide is released. At present such experiments are challenging because the natural redox partner of FadE has not been identified. This study supports the hypothesis that the degree of internal ROS production can depend upon the type of active metabolism inside cells.

## Introduction

The discoveries of catalase and of superoxide dismutase were adventitious: their discoverers had no reason to expect that such enzymes would exist, because their substrates, $H_2O_2$ and $O_2^-$, were not known to be involved in biology [1, 2]. However, the investigators quickly inferred that these oxygen species must arise inside cells and be capable of damaging the organism. This prediction proved to be correct. In 1986 Carlioz and Touati engineered a strain

Ladkrabang (No. 2566-02-16-001) to C.S. and NIH grant GM49640 To J.A.I.

**Competing interests:** The authors have declared that no competing interests exist.

of *E. coli* that lacked its two cytoplasmic superoxide dismutases [3]. The mutant grew at normal rates under anaerobic conditions, but it exhibited severe defects in oxic medium. Similar results were subsequently obtained using mutants that lacked both catalase and NADH peroxidase (AhpCF), the major scavengers of hydrogen peroxide [4].

The growth defects of both strains included an inability to synthesize aromatic and branched-chain amino acids, plus a failure to catabolize non-fermentable carbon sources. These defects arose because both $O_2^-$ and $H_2O_2$ disable enzymes that have solvent-exposed iron cofactors [5]. These species oxidize the metals, triggering their dissociation from the enzyme and the loss of activity. Such enzymes exist in each of the vulnerable pathways.

The existence of these phenotypes confirmed that $O_2^-$ and $H_2O_2$ are formed inside aerobic cells. The efflux of $H_2O_2$ from catalase/peroxidase-deficient cells (*katG katE ahpCF*) was subsequently quantified, revealing that $H_2O_2$ arises inside air-saturated, glucose-fed *E. coli* at a rate of 10–15 µM/s [6]. A subset of this $H_2O_2$ presumably is a by-product of the formation of $O_2^-$, which is then dismutated to $H_2O_2$.

However, it is not understood how these reactive oxygen species are generated. The rate of intracellular $H_2O_2$ production rises in rough proportion to the oxygen concentration; these first-order kinetics contrast with the saturable Michaelis-Menten kinetics that would be expected if $H_2O_2$ were generated by an oxygen-binding enzyme [7]. Thus, workers presume that these reactive oxygen species (ROS) are generated when oxygen accidentally collides with an electron donor and chemically oxidizes it. Indeed, hyperoxia imposes upon wild-type cells many of the same phenotypes that were exhibited under normoxia by scavenger-less mutants [8].

Although oxygen is broadly regarded as a strong oxidant, its chemical reduction is not easy. Molecular oxygen is a diradical, with two unpaired electrons in spin-aligned orbitals (Fig 1). In contrast, most organic molecules contain spin-paired electrons. The rules of quantum mechanics dictate that electron transfer to the diradical species can occur only in single-electron steps [9]. This restriction greatly suppresses the reactivity of oxygen, because its affinity for the first electron is low: Eo' = −0.16 V. For this reason, molecular oxygen cannot spontaneously oxidize the amino acids, lipids, carbohydrates, and nucleic acids that constitute the bulk of cell material. Instead, electron transfer to oxygen requires a good univalent electron donor. Inside cells univalent redox reactions are mediated by flavins, metal centers, and quinones, and the search for ROS sources has therefore focused upon them.

Soon after the discovery of superoxide dismutase, Vince Massey and colleagues reported that flavoenzymes can adventitiously transfer electrons to oxygen, generating superoxide [10]. The rates at which they did so varied over orders of magnitude, indicating that structural or electronic features of the enzyme have a strong influence upon this behavior of the flavin. Soon after, studies of the mitochondrial respiratory chain revealed that if the catalytic cycle of the bc1 complex is stalled by inhibitors, its semiquinone intermediates also leak electrons to oxygen [11]. Both findings supported the prediction that these flavin and quinone cofactors could be sources of ROS.

Efforts have been made to pinpoint the main contributors inside living cells. Because the respiratory chain features a large electron flux through flavins, quinones, and metal centers, respiration was expected to be the primary source of ROS in aerobic cells. Indeed, respiring membrane vesicles from *E. coli* release small amounts of $O_2^-$ and $H_2O_2$. Dissection of the chain pinpointed two primary sites of leakage. The first was NADH dehydrogenase II, a simple enzyme whose FAD cofactor receives a hydride anion from NADH and then normally transfers the electrons consecutively to a bound molecule of ubiquinone [12]. The $FADH_2$ moiety turned out to be the site of accidental transfer to oxygen. This reaction generated superoxide, but most of the nascent superoxide immediately abstracted the second electron from the flavosemiquinone, so that more $H_2O_2$ than $O_2^-$ exited the active site.

**Fig 1. Flavin oxidation can be an intracellular source of both O2- and $H_2O_2$.** Molecular oxygen can abstract a single electron from dihydroflavin, forming a caged pair of $O_2^-$ and flavosemiquinone. If $O_2^-$ dissociates (left pathway), a second oxygen molecule will accept the remaining electron, yielding a second molecule of $O_2^-$. Cellular superoxide dismutase converts these molecules to $H_2O_2$. Alternatively, after a spin inversion (right pathway), $O_2^-$ can combine with flavosemiquinone to form a peroxy adduct, which upon hydrolysis releases $H_2O_2$.

A second source of ROS proved to be fumarate reductase [13]. The enzyme is synthesized under anoxic conditions, as its role is to transfer electrons received from quinones travel through three integral iron-sulfur clusters to a superficial flavin, which then transfers a hydride anion to fumarate. The enzyme may encounter oxygen when erstwhile anaerobic cells enter an oxic habitat. The study showed that when oxygen collides with the reduced $FADH_2$, a mixture of $O_2^-$ and $H_2O_2$ is formed. Growth studies confirmed that fumarate reductase is the predominant source of both $O_2^-$ and $H_2O_2$ in the special situation of bacterial movement into an oxic environment [14].

However, the respiratory chain appears not to be the major source of ROS inside cells that are steadily aerated. Deletion of the NADH dehydrogenases, which deliver electrons into the chain, eliminated ROS production by membrane vesicles but had no apparent impact upon overall $H_2O_2$ formation by whole cells [15]. Some intracellular $H_2O_2$ is known to be produced by aspartate oxidase, a fumarate reductase homolog that initiates NAD biosynthetic pathway, and a minor amount is generated by the oxidation of the menaquinone pool [16]. But the majority of the $H_2O_2$ produced in *E. coli* is unaccounted for. This remains one of the major unanswered questions in the field of oxidative stress.

In this study we examined whether acyl-CoA dehydrogenase, a flavoenzyme integral to fatty acid catabolism, might be predisposed to autoxidation. This enzyme is believed to transfer electrons from fatty acyl-CoA substrates to a putative electron-transfer flavoprotein, which then delivers them to the respiratory chain [17, 18]. We observed that when the dehydrogenase, encoded by FadE, is strongly expressed, it elevates the rate of cellular $H_2O_2$ production. This result contributes to the consensus that flavoenzymes are likely sites of ROS formation in vivo, and it suggests that cells may experience higher levels of oxidative stress when their catabolic strategy involves high electron flux through such enzymes.

## Experimental procedures

### Bacterial strains, and growth conditions

Bacterial strains and oligonucleotide primers used in this study are indicated in Table 1. Mutant alleles were transferred to strains by standard P1 transduction [19]. Transductants

**Table 1. Bacterial strains, and oligonucleotide primers.**

| bacterial strains or Primers | General description or relevant genotype | References |
|---|---|---|
| *E. coli* **Strains:** | | |
| NEB 5-alpha | *fhuA2 (argF-lacZ)U169 phoA glnV44 80 (lacZ)M15 gyrA96 recA1 relA1 endA1 thi-1 hsdR17* | New England Biolabs |
| MG1655 | Wild type *E. coli* K-12 | Lab stock |
| JWC268 | Δ*fadE* of MG1655 (Kan$^R$) | Lab stock |
| 381 RW11 | Δ*fadR::Tn10* (Tet$^R$) | Lab stock |
| AL441 | As MG1655 *ΔlacZ1 attλ::[pSJ501:: katG'-lacZ$^+$]~cat* (Chl$^R$) | [21] |
| AL495 (HPX$^-$) | *Δ(katG17::Tn10)1 (ahpC-ahpF')del kan::'ahpF Δ(katE12::Tn10)1 ΔlacZ1 attλ::[pSJ501:: katG'-lacZ$^+$]~cat* (Kan$^R$, Chl$^R$) | [49] |
| Δ*fadE* AL441 | *ΔfadE, ΔlacZ1 attλ::[pSJ501:: katG'-lacZ$^+$]~cat* (Kan$^R$, Chl$^R$) | P1vir(JWC268)×AL441 |
| Δ*fadR* AL441 | *ΔfadR, ΔlacZ1 attλ::[pSJ501:: katG'-lacZ$^+$]~cat* (Tet$^R$, Chl$^R$) | P1vir(381 RW11)×AL441 |
| Δ*RE* AL441 | *ΔfadR, ΔfadE, ΔlacZ1 attλ::[pSJ501:: katG'-lacZ$^+$]~cat* (Tet$^R$, Kan$^R$, Chl$^R$) | P1vir(381 RW11)× Δ*fadE* AL441 |
| AL427 | *Δ(katG17::Tn10)1 Δ(ahpCF1::cat)1 Δ(katE12::Tn10)1* (Tet$^s$, Kan$^s$, Chl$^s$) | [49] |
| Δ*fadR* AL495 | *ΔfadR, Δ(katG17::Tn10)1 (ahpC-ahpF')del kan::'ahpF Δ(katE12::Tn10)1, ΔlacZ1 attλ::[pSJ501:: katG'-lacZ$^+$]~cat* (Tet$^R$, Kan$^R$, Chl$^R$) | P1vir(381 RW11)×AL495 |
| Δ*fadE* AL427 | *ΔfadE, Δ(katG17::Tn10)1 Δ(ahpCF1::cat)1 Δ(katE12::Tn10)1* (Kan$^R$) | P1vir(JWC268)×AL427 |
| Δ*fadE* SK/O | *ΔfadE, Δ(katG17::Tn10)1 Δ(ahpCF1::cat)1 Δ(katE12::Tn10)1 ΔlacZ1 attλ::[pSJ501:: katG'-lacZ$^+$]~cat* (Kan$^R$, Chl$^R$) | P1vir(AL441)× Δ*fadE* AL427 |
| Δ*RE* SK/O | *ΔfadR, ΔfadE, Δ(katG17::Tn10)1 Δ(ahpCF1::cat)1 Δ(katE12::Tn10)1 ΔlacZ1 attλ::[pSJ501:: katG'-lacZ$^+$]~cat* (Kan$^R$, Chl$^R$, Tet$^R$) | P1vir(381 RW11)× Δ*fadE* SK/O |
| **Primers** | | |
| +200 FadE FP | 5' GTG TAC CGG ATA CCG CCA AA 3' | This study |
| -200 FadE RP | 5' TGA CGG GGC TGT TCT CG 3' | This study |
| FadR-KOFP+161 | 5' AAC GGT CAG GCA GGA 3' | This study |
| FadR-KORP-200 | 5' ATA ATC GCG CAC CGC 3' | This study |

were confirmed by colony PCR (see Table 1). During constructions, bacterial strains were routinely maintained in Luria-Bertani (LB) broth [20] or agar supplemented with the appropriate antibiotics. Experimental media used M9 salts [20], 0.2% casamino acids, 0.5 mM tryptophan, and 1 μg/ml thiamine, and contained 0.4% glycerol and/or 50 μM dodecanoic acid as carbon sources, as indicated. A 50 mM stock solution of dodecanoic acid (Sigma) was prepared in ethanol before dilution into the working media. The final ethanol concentration (0.1%) did not affect bacterial growth.

### ß-Galactosidase reporter assay

Whether FadE makes enough $H_2O_2$ to activate the OxyR regulon was tested using a *katG-lacZ* transcriptional fusion. AL441 is a derivative of the wild-type strain MG1655; it includes the fusion plus a deletion of the native *katG* allele. This and its derivative *fadR* and *fadE* strains were cultured overnight in M9 medium containing glycerol. The resultant stationary phase cultures were diluted to 1% into M9 medium supplemented with 0.2% casamino acids, 0.5 mM tryptophan, and 1 μg/ml thiamine. The cultures were grown with shaking at 37 C to an $OD_{600}$ of approximately 0.2. The cells were collected by centrifugation, washed twice with M9 salt solution, and subsequently diluted to $OD_{600}$ of about 0.025 in the M9 based media as described above with various carbon sources (glycerol, glycerol + dodecanoate, or only dodecanoate). The cultures were grown at 37 C with shaking for two to three generations. ß-Galactosidase activity was then examined. The cells were harvested through centrifugation, washed with phosphate-buffered saline (PBS), resuspended in Z buffer, lysed with lysozyme, and assayed for β-galactosidase activity at 420 nm by spectrophotometer [20]. The data were collected from five independent colonies with two technical replicates apiece.

### Measurement of intracellular hydrogen peroxide production

To quantify the production of intracellular hydrogen peroxide production, the scavenging-deficient *E. coli* strain (AL495, HPX⁻ strain) was used as a parental strain [21]. Deletion alleles of *fadR* and *fadE* were introduced to the AL495 strain by standard P1 transduction procedure [19] and subsequently confirmed by colony PCR. Primers specific to each gene are indicated in Table 1. The level of hydrogen peroxide released by sterile culture medium was also examined. All bacterial cells were initially streaked on standard M9 medium (CSH protocol) with glucose in a Coy anaerobic chamber. Overnight cultures were then grown in the same medium in the anaerobic environment. Cells were then directly inoculated at 1% dilution into oxic M9 glycerol and dodecanoate media supplemented with 0.2% casamino acids, 0.5 mM tryptophan, and 1 μg/ml thiamine, and this preculture was grown to $OD_{600}$ approximately 0.2. These log-phase cells were collected by centrifugation at 4000 x g for 5 min. at room temperature, resuspended in the same fresh media, and finally diluted into the same fresh media to $OD_{600}$ at an $OD_{600}$ of approximately 0.02. These cultures were further incubated with shaking at 37C. At selected time points, 1-ml aliquots were removed, the cells were pelleted by 1 min. centrifugation in a microfuge, and the hydrogen peroxide content of the supernatant was examined by the Amplex Red/horseradish peroxidase method [22]. Fluorescence was measured in a Shimadzu RF Mini-150 fluorometer and converted to $H_2O_2$ concentration using a curve obtained from standard samples in the same assay medium. The secondary growth of various *E. coli* strains at the same selected time point was also examined over 42 min. The mean results were calculated from four independent biological samples. As in previous studies, the levels of $H_2O_2$ production were undetectable in wild-type cells containing catalase and peroxidase.

## Bacterial growth kinetics

The growth of wild-type, $\Delta fadR$, HPX$^-$, and HPX$^-$ $\Delta fadR$ derivatives of MG1655 were determined by spectrophotometer at 600 nm. The culture conditions were equivalent to those detailed in measurements of $H_2O_2$ production.

## Reactivity of dihydrolipoic acid with hydrogen peroxide in vitro

The ability of reduced lipoic acid (dihydrolipoate) to scavenge $H_2O_2$ was measured in vitro. The total reaction (2 ml) contained 50 mM sodium phosphate buffer, pH 7.4, 1 mM of the iron chelator diethylenetriaminpentaacetic acid (DTPA), 10 μM $H_2O_2$, and 100 μl of sample. The DTPA was included to avoid any iron-catalyzed chemistry. At timepoints 50 μl of 0.25 mM Amplex UltraRed (AR) and 100 μl of 0.25 mg/ml HRP (Sigma P8250) were added. Reactions were fast, and the final fluorescence was examined after 20 seconds by fluorometer (Shimadzu model RF-mini150), using 520 nm and 620 nm filters for excitation and emission wavelengths, respectively [22]. The concentration of hydrogen peroxide was quantified using a standard curve with known amount of $H_2O_2$. Dithiothreitol (DTT) was used as a positive control, as its thiol groups are known to react with $H_2O_2$.

## Results and discussion

Flavoenzymes have repeatedly been observed to transfer electrons from their reduced flavins to molecular oxygen, generating a mixture of superoxide and $H_2O_2$ [5]. This observation makes good chemical sense, as flavins are good univalent electron donors, and they are typically situated at the enzyme surface where collisions with dissolved oxygen are likely. However, the contribution of any particular flavoenzyme to the net ROS production inside the cell depends additionally on the titer of the enzyme. Central metabolism involves much higher substrate fluxes than do other pathways, and accordingly its enzymes are abundant. Therefore, it seemed plausible that FadE, an enzyme which participates in the degradation of fatty acids as a main carbon source, could make a substantial contribution to endogenous ROS production. The placement of FadE in this process is shown in Fig 2.

The rate of internal $H_2O_2$ production can be directly measured using *E. coli* mutants that lack $H_2O_2$-scavenging enzymes [15]. Any $H_2O_2$ formed inside the cell flows across the cell membrane and into the culture medium, where its progressive accumulation can be measured. The external $H_2O_2$ concentration eventually reaches a steady-state level, as there is slow residual scavenging by the cytochrome oxidases of the cell, which accept $H_2O_2$ as a poor pseudosubstrate that competes with molecular oxygen [23]. Both the initial rate of $H_2O_2$ efflux, and the final steady state, are good measures of the rate of its internal production.

The rates of $H_2O_2$ formation were measured when cells were fed a mixture of glycerol and dodecanoic acid. Dodecanoic acid is a soluble substrate for the fatty acid degradation pathway; however, it cannot serve as sole carbon source for Hpx$^-$ strains, because the acetyl-CoA produced by this process is inefficiently catabolized by the TCA cycle of these cells, due to the $H_2O_2$ sensitivity of aconitase and fumarase. Glycerol can be degraded independently of the TCA cycle, and unlike glucose its presence does not shut down expression of the fatty acid catabolic genes.

We observed that $H_2O_2$ production was substantially elevated in a *fadR* mutant, which strongly induces fatty-acid degrading enzymes [18, 24], including FadE (Fig 3). The rate returned to that of a FadR$^+$ strain when *fadE* was absent. These data show that FadE can be a predominant source of $H_2O_2$ when it is fully induced.

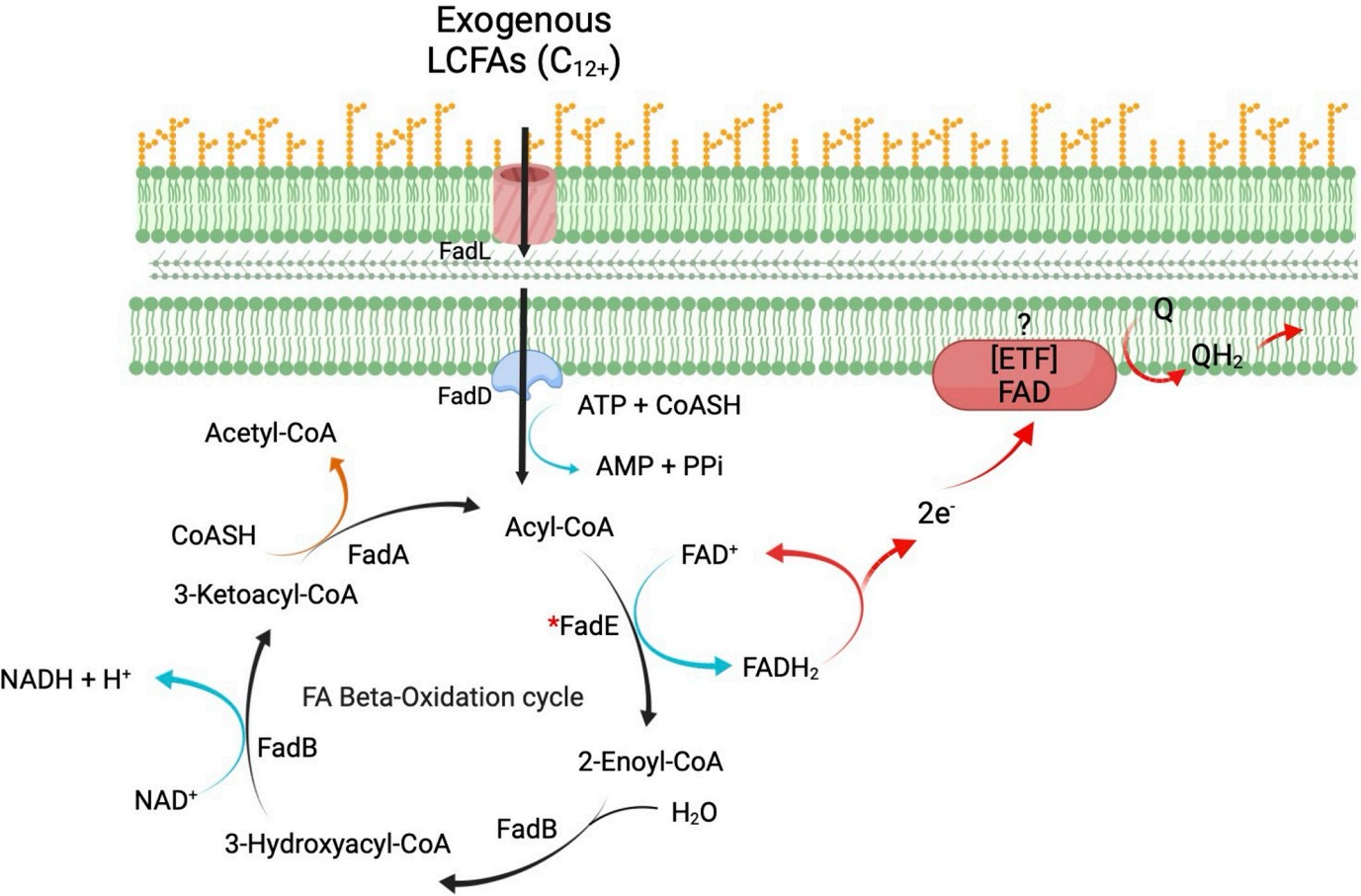

**Fig 2. Overview of the pathway of long-chain fatty acid degradation.** FadE (acyl-CoA dehydrogenase) is marked with an asterisk. In this figure it transfers two electrons to the quinone pool via Electron Transfer Flavoprotein (ETF). However, it is possible that ETF is absent from *E. coli* and FadE is a membrane protein that reduces quinones directly.

## The FadE-expressing strain exhibits poor growth in the Hpx⁻ background

Because $H_2O_2$ damages key enzymes in amino-acid biosynthetic pathways, casamino acids were supplemented in the growth medium. Nevertheless, an Hpx⁻ strain grows slightly more slowly than scavenging-proficient cells. That growth defect was substantially worse for the *fadR* mutant (Fig 4). The defect was especially pronounced when cells were resuspended at low cell density, when $H_2O_2$ clearance by respiration in minimized. This defect was ameliorated in a strain lacking FadE. The *fadR* mutation had little effect upon growth in a scavenging-proficient strain. Thus, the growth phenotype supported the notion that FadE can be an important source of $H_2O_2$.

## Scavenging enzymes keep FadE-derived $H_2O_2$ at non-threatening levels

*E. coli* uses the OxyR transcription factor to sense high levels of $H_2O_2$ that might debilitate the cell [25, 26]. When activated, OxyR induces a handful of defensive systems, including catalase G, encoded by *katG*. Workers have relied upon *katG-lacZ* transcriptional fusions as sensitive reporters of $H_2O_2$ stress. These fusions are induced > ten-fold in glucose-grown Hpx⁻ strains,

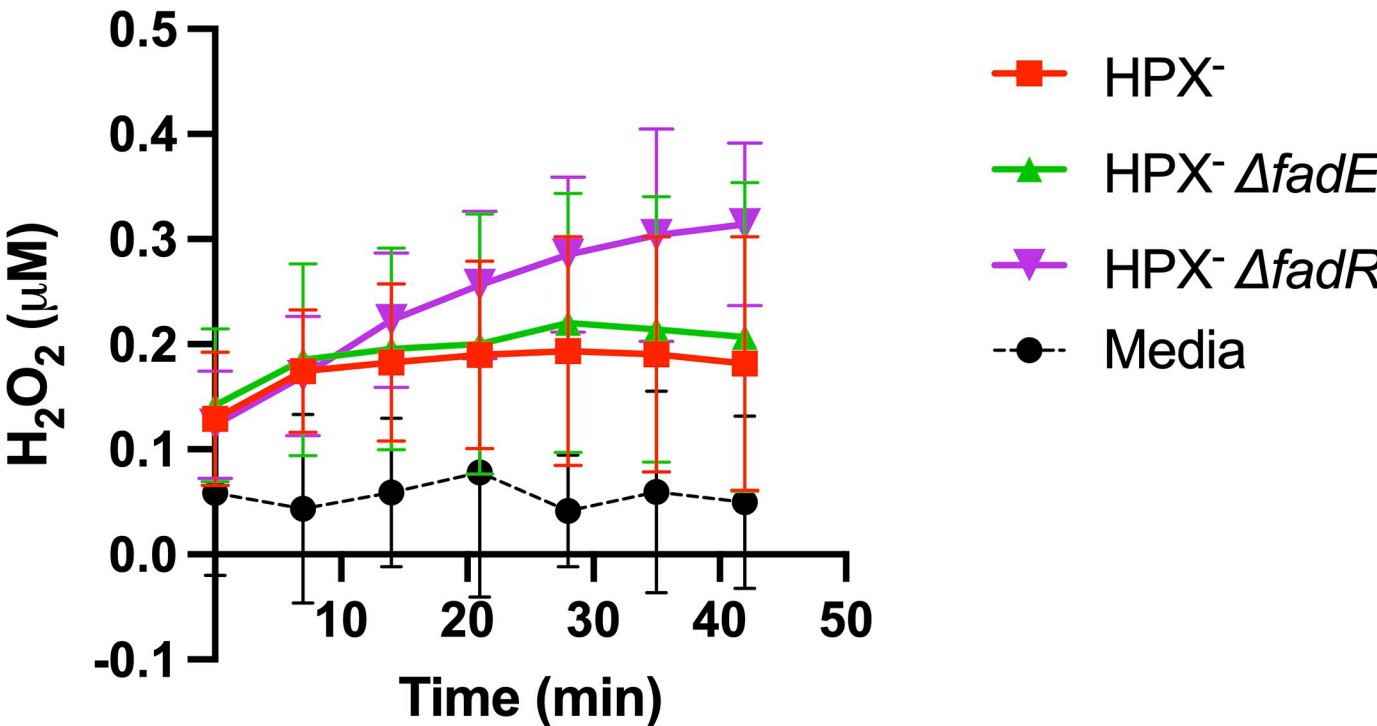

**Fig 3. FadE-dependent H₂O₂ production by whole cells.** Mutants lacking the $H_2O_2$-scavenging enzymes catalase and peroxidase (HPX⁻) were suspended in medium containing dodecanoic acid, and $H_2O_2$ release was monitored. The *fadR* mutant fully induces the fatty-acid degradation pathway, including FadE.

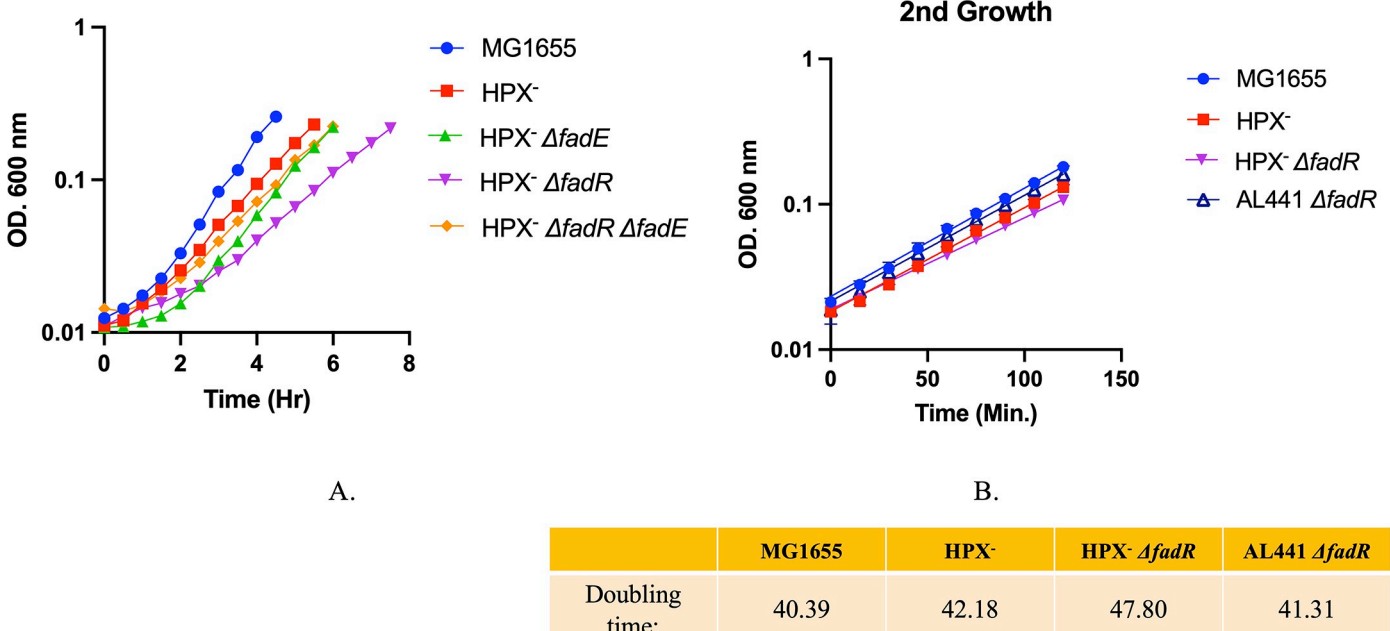

A.

B.

| | MG1655 | HPX⁻ | HPX⁻ *ΔfadR* | AL441 *ΔfadR* |
|---|---|---|---|---|
| Doubling time: | 40.39 | 42.18 | 47.80 | 41.31 |

**Fig 4. Non-scavenging mutants grow poorly when catabolizing dodecanoic acid.** (A). At time zero cells were shifted to medium containing glycerol and dodecanoic acid. (B) Even after adaptation to the medium, the *fadR* mutant exhibited slower growth.

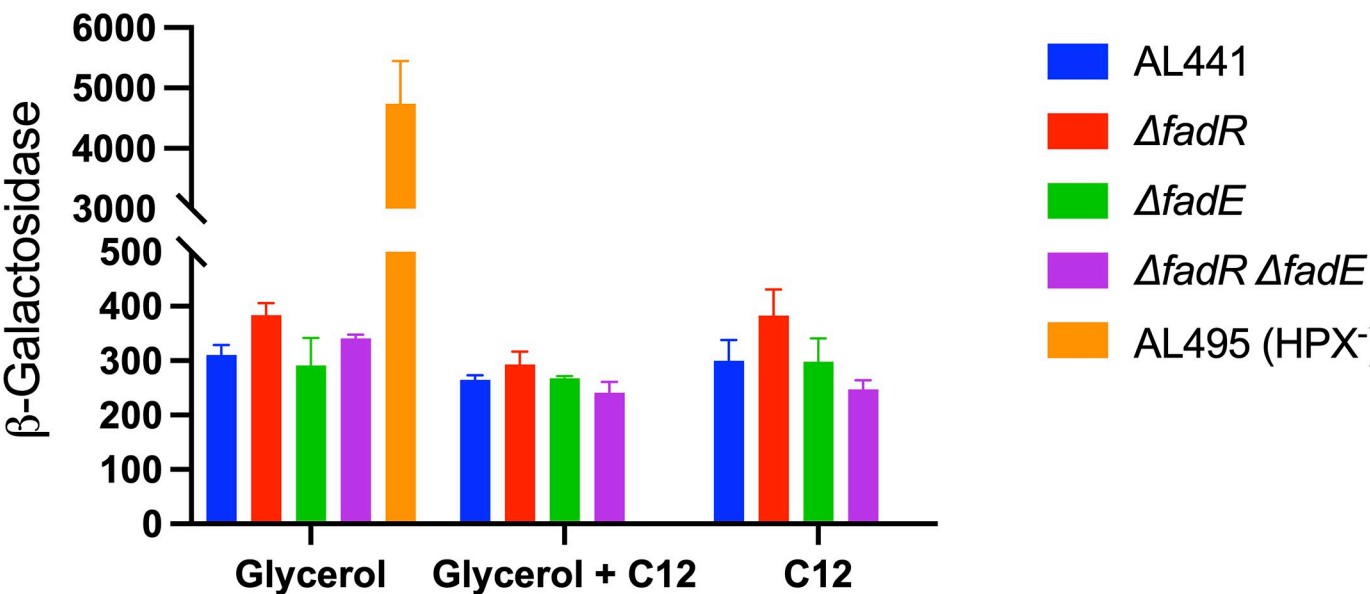

**Fig 5. Beta-oxidation does not generate enough H₂O₂ to induce the OxyR regulon.** Scavenging-proficient cells were cultured with glycerol and/or dodecanoic acid as a carbon source, and the expression of an OxyR-controlled *katG-lacZ* fusion was monitored. Expression in a catalase/peroxidase-deficient mutant (HPX⁻) is shown for comparison.

whose 1 micromolar intracellular $H_2O_2$ is much higher than the 50 nM level inside wild-type cells [27].

The transcriptional fusion was inserted into a scavenger-competent (catalase/peroxidase-proficient) strain. Mutations in *fadR* and/or *fadE* were introduced. When these strains were cultured in media containing glycerol and/or dodecanoate, the level of *katG-lacZ* expression was similar (Fig 5). It appears that basal levels of catalase and peroxidase are sufficient to protect the cell from any $H_2O_2$ generated by FadE.

## Lipoic acid does not degrade $H_2O_2$

Fatty acid catabolism requires high-level expression of enzymes of the TCA cycle. Because 2-oxoglutarate dehydrogenase requires lipoic acid as a cofactor, lipoic acid is substantially synthesized during periods of fatty acid degradation. Other workers have reported that the reduced form of lipoic acid, dihydrolipoamide, has antioxidant properties, including the abilities to scavenge $O_2^-$ [28], to chelate metal ions [29], and to regenerate other scavenging biomolecules such as glutathione, vitamin C, and vitamin E [29–31]. There was some disagreement as to whether lipoate might also degrade $H_2O_2$; the presence of its dithiol moieties made this plausible.

This idea was tested by direct measurements of $H_2O_2$ during incubation with dihydrolipoate. Low-micromolar concentrations were used, in contrast to previous experiments, to more closely mimic physiological doses. Fig 6 shows that dihydrolipoate was ineffective at $H_2O_2$ degradation, in contrast to dithiothreitol, a dithiol whose ability to degrade $H_2O_2$ has been established [21]. The difference is surprising, given that the reported reduction potentials of dithiothreitol (-0.33 V) and lipoic acid (-0.29 to -0.33 V) are similar [31]. In any case, lipoate synthesis is unlikely to protect cells from oxidants formed during fatty acid catabolism.

The major sources of endogenous ROS in bacteria are unclear. This study indicates that FadE activity promotes $H_2O_2$ production inside cells. The simplest explanation would be that

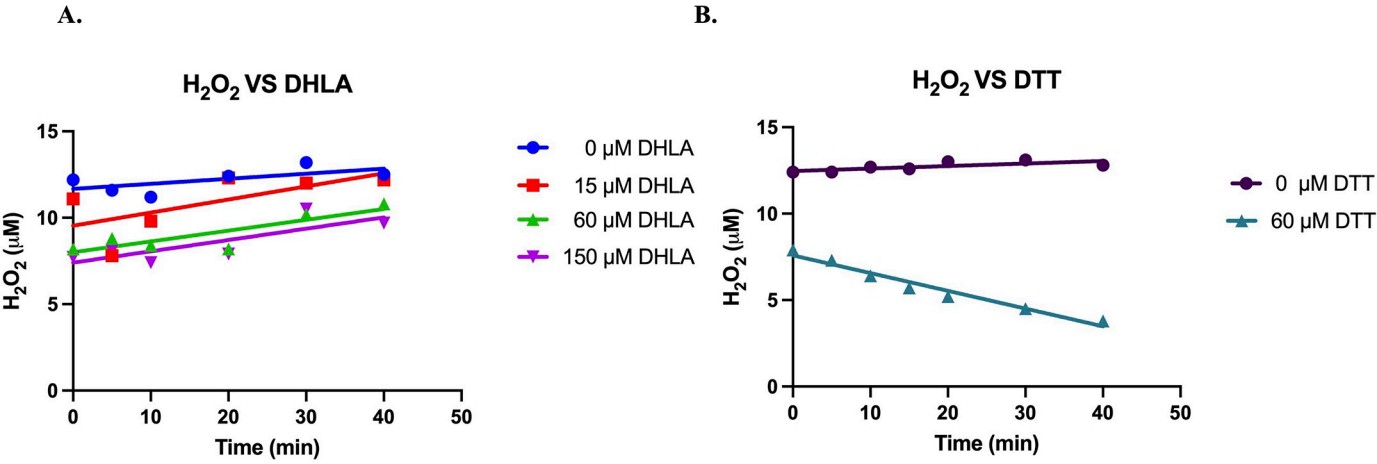

**Fig 6. Dihydrolipoic acid (DHLA) does not degrade H$_2$O$_2$.** Scavenging by dithiothreitol (DTT) is shown for comparison (B).

the reduced FADH$_2$ intermediate of FadE itself inadvertently transfers some electrons directly to oxygen, likely producing a mixture of superoxide and H$_2$O$_2$. An alternative model, suggested by others, is that the catabolism of fatty acids increases the flux of NADH to the standard NADH dehydrogenase-initiated respiratory chain, and that electron leakage from the chain will elevate cellular ROS [32]. The present data do not rule out this second model; however, our prior work indicated that electron leakage from the components of the normal aerobic chain is minimal [15].

The FadE enzyme belongs to the general family of acyl-CoA dehydrogenases (ACDs). These enzymes are ubiquitous in the biota and are invariably required for fatty acid catabolism. In some organisms, including humans, other ACD family members participate in the oxidative degradation of leucine, valine, isoleucine, lysine, and tryptophan. In each case the ACD accomplishes the α,β-desaturation of CoA thioesters. In well-characterized mitochondrial enzymes, once the ACD FAD cofactor receives a hydride anion from substrate, it then consecutively transfers the two electrons to the flavin of a bound electron transfer flavoprotein partner (ETF); in turn, ETF transfers the electrons to the electron-transport chain. A single mitochondrial ETF protein acts as a hub that accepts electrons from a variety of ACD family members [33]. Mechanistic details of ACDs and their interactions with ETF have been reported [34].

## Electron transfer to oxygen by eukaryotic ACDs

The physical structures of eukaryotic ACDs have evolved to suppress electron leakage to oxygen. The reduced FADH$_2$ cofactor is largely buried in polypeptide, and electron transfer to its proper substrate, ETF, occurs over distance by a typical electron hopping event. In biochemical assays ETF can be replaced by high-potential artificial electron acceptors. The radical status of molecular oxygen means that in principle it is similarly capable of receiving electrons in this way; however, its univalent potential is low, and this quality, plus its inability to closely approach the flavin, ensure oxidase activity is minimal. Interestingly, investigators have found that ACD releases electrons to oxygen more easily if the *trans*-2-enoyl-CoA substrate is absent, indicating that in this circumstance molecular oxygen gains more access to the FADH$_2$ by entering the substrate cavity; in this view its proximity compensates somewhat for its lower

potential. The usual catalytic cycle, however, ensures that electrons move to ETF before product dissociates—thereby avoiding ROS formation.

This analysis is supported by the contrary behavior of acyl-CoA oxidases, a related family of enzymes in which acyl-CoA dehydrogenation proceeds as with ACDs but is followed by stoichiometric electron transfer to molecular oxygen. The product is $H_2O_2$, and so these enzymes are localized in catalase-filled peroxisomes. These enzymes do not bind ETF, and their high turnover number to oxygen is enabled by markedly looser polypeptide packing around the flavin, providing greater exposure to solvent [35]. A similar effect has been achieved by mutagenesis that relieves protein packing around the ACD flavin [36]. These observations all show that ACD flavins are chemically capable of generating some level of ROS, but that protein structure can suppress it.

It is in this context that *E. coli* FadE should be considered. Unlike eukaryotic cells, while *E. coli* oxidizes fatty acids, it does not catabolize branched-chain amino acids, and it degrades lysine and tryptophan by other routes. The ETF-hub model therefore does not apply, and in fact no ETF partner to FadE has been identified in *E. coli*, allowing the possibility that none is used and that FadE associates directly with the chain [18]. Indeed, protein-localization prediction programs have suggested that *E. coli* FadE may itself be embedded in the cytoplasmic membrane, although this prediction is not unanimous [37]. Therefore, it is unclear whether the key structural and kinetic features of the eukaryotic enzymes—including their devices to avoid ROS production—also pertain to this bacterial enzyme.

Unfortunately, *E. coli* FadE has been the subject of only limited biochemical investigation [18, 17, 38]. Alphafold [39, 40] predicts the structure shown in Fig 7A. Analysis by the MMseqs2 algorithm [41] reports that the *E. coli fadE* shows the high similarity to acyl-CoA oxidase 1 (ACOX-1) from *C. elegans* (AFDB accession number: AF-O62140-F1). The structure of ACOX-1 with bound FAD and ATP (PDB: 5K3I) presents as a homodimer. Like all other members of the acyl-CoA dehydrogenase family, it contains a catalytic glutamate residue to install the double bond at α-β position, generating enoyl-CoA as a product (Fig 7B) [34, 35].

**A.**                                                                    **B.**

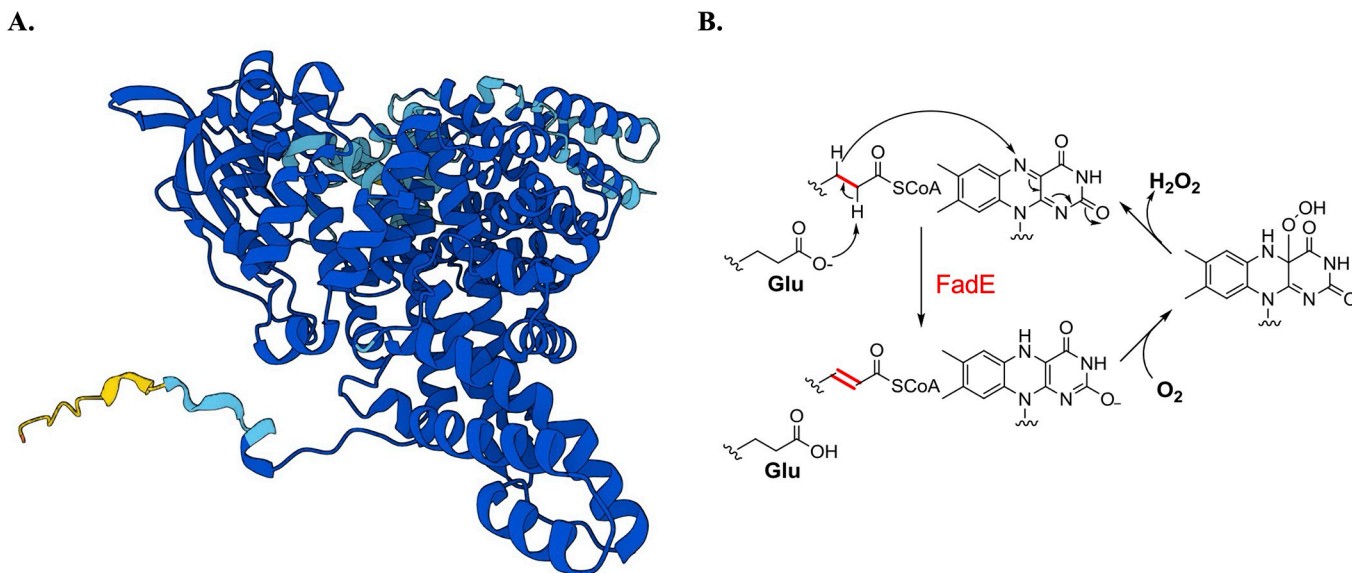

**Fig 7. FadE structure and possible route of $H_2O_2$ production.** (A) Structure predicted by AlphaFold. (B) Predicted oxidation mechanism. Proton abstraction by a glutamate residue triggers transfer of a hydride anion to the adjacent FAD. The subsequent oxidation of FADH₂ would proceed by either of the two routes depicted in Fig 1.

The FAD cofactor lies at the interface of dimeric enzyme [42], enabling its exposure to the solvent and the direct generation of hydrogen peroxide. A structural similarity of FadE to ACOX-1 may support the possibility that this ACD generates ROS to a greater extent than eukaryotic ACDs. In vitro studies will be needed to resolve this point—but they will require that the presence or absence of an ETF first be settled.

## The physiological impact of a new ROS source

Hydrogen peroxide is not the sole ROS species likely to be produced during any oxidation of flavoproteins. Because molecular oxygen is constrained to accept electrons in univalent steps, dihydroflavin autoxidation produces a molecule of superoxide as the immediate product. The superoxide molecule can either exit the active site or else recombine with the adjacent flavosemiquinone to form a peroxy adduct [43]. This flavin species spontaneously hydrolyzes, yielding $H_2O_2$ and oxidized FAD. Thus, $FADH_2$ oxidation can generate a mixture of $O_2^-$ and $H_2O_2$. In enzymes that lack other redox moieties, such as aspartate oxidase and NADH dehydrogenase II, virtually no $O_2^-$ leaves the active site, indicating that the peroxide route is favored. However, xanthine oxidase and fumarate reductase possess metal centers proximate to the flavin and produce much more superoxide to the bulk solution [13, 44]. Analysis suggests that the electron of the flavosemiquinone shifts to those redox centers rather than recombining with superoxide. If FadE associates with an ETF, then a similar mechanism might enable its flavin to generate intracellular $O_2^-$; if it does not, then $H_2O_2$ may be the predominant product.

The question of whether FadE releases superoxide to the cell interior carries physiological import. When grown in glucose, *E. coli* calibrates its titer of SOD so it is barely sufficient to suppress enzyme damage by endogenous superoxide [45]. Therefore, if the production of endogenous superoxide is greater when cells consume fatty acids, the threshold for overt injury would be approached. Two observations raise the possibility that *E. coli* may be buffered against this situation. First, the transcription of *fadE* is inhibited by phospho-ArcA, the form of the ArcA transcription factor that accumulates when the quinone pool is reduced [46]. Phospho-ArcA also inhibit transcription of *sodA*, encoding manganese-cofactored SOD [47]. This parallel raises the possibility that fatty acid degradation occurs concomitant with some level of SOD induction.

Second, aconitase of the TCA cycle is acutely sensitive to $O_2^-$. Gardner and Fridovich pointed out that if excess $O_2^-$ is generated by TCA cycle-dependent processes, then the inactivation of aconitase might act as a circuit-breaker [48]. In the current example, excess $O_2^-$ production by FadE might inactivate aconitase. The acetyl-CoA produced by fatty acid oxidation would then accumulate, and further fatty acid catabolism would stall for want of free CoA. In this speculative scheme, FadE-driven $O_2^-$ formation would be suppressed until sufficient SOD were present to protect superoxide-sensitive enzymes.

The example of FadE raises the possibility that the degree of endogenous ROS stress depends upon the foodstuff and the metabolic strategy of the bacterium. The present data do not adequately test this point. Biochemical characterization of FadE, including reconstitution of its full catalytic cycle in vitro, will be needed.

## Supporting information

**S1 File. The tables show raw data of all experiments above.**
(DOCX)

## Acknowledgments

We are grateful to John Cronan for providing guidance and resources throughout this investigation.

## Author Contributions

**Conceptualization:** Chaiyos Sirithanakorn, James A. Imlay.

**Data curation:** Chaiyos Sirithanakorn, James A. Imlay.

**Formal analysis:** Chaiyos Sirithanakorn.

**Funding acquisition:** Chaiyos Sirithanakorn, James A. Imlay.

**Investigation:** Chaiyos Sirithanakorn, James A. Imlay.

**Methodology:** Chaiyos Sirithanakorn.

**Project administration:** Chaiyos Sirithanakorn.

**Resources:** Chaiyos Sirithanakorn.

**Software:** Chaiyos Sirithanakorn.

**Supervision:** Chaiyos Sirithanakorn, James A. Imlay.

**Validation:** Chaiyos Sirithanakorn.

**Visualization:** Chaiyos Sirithanakorn, James A. Imlay.

**Writing – original draft:** Chaiyos Sirithanakorn.

**Writing – review & editing:** Chaiyos Sirithanakorn, James A. Imlay.

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
