## [Decision Letter · Decision Letter 0]

30 Jul 2024

PONE-D-24-26458Evidence for endogenous hydrogen peroxide production by E. coli fatty acyl-CoA dehydrogenasePLOS ONE

Dear Dr. Sirithanakorn,

Thank you for submitting your manuscript to PLOS ONE. After careful consideration, we feel that it has merit but does not fully meet PLOS ONE’s publication criteria as it currently stands. Therefore, we invite you to submit a revised version of the manuscript that addresses the points raised during the review process.

We look forward to receiving your revised manuscript.

Kind regards,

Vipin Chandra Kalia, FNASc, FAMI

Academic Editor

PLOS ONE

Journal Requirements:

This study was financially supported by King Mongkut's Institute of Technology Ladkrabang (No. 2566-02-16-001) to C.S. and NIH grant GM49640 To J.A.I.

We are grateful to John Cronan for providing guidance and resources throughout this investigation. This study was financially supported by King Mongkut's Institute of Technology Ladkrabang (No. 2566-02-16-001) to C.S. and NIH grant GM49640 To J.A.I

This study was financially supported by King Mongkut's Institute of Technology Ladkrabang (No. 2566-02-16-001) to C.S. and NIH grant GM49640 To J.A.I.

5. We note that your Data Availability Statement is currently as follows: All relevant data are within the manuscript and its Supporting Information files.

**Additional Editor Comments:**

Reviewers have reported that the study is interesting and needs minor revision.

You are requested to meticulously and carefully respond to the reviewers comments.

Reviewer's Comments 1:

Review comment on “Evidence for endogenous hydrogen peroxide production by E. coli fatty acyl-CoA dehydrogenase”

General comment: This is a nice piece of work on detail understanding on how superoxides and H2O2 are produced continuously in bacterial cells. Results presented in figures but not described elaborately, indeed results are given partially as in typical “results & Discussion” section, while this article has again a separate discussion section. So, it is actually better combine both sections as “Results and Discussion”, and describe a little of the Figures as a result component of the section.

Specific comment:

Lines22-23: “To avoid this problem cells maintain high levels of scavenging enzymes” provide names of a few important scavenging enzymes relevant to this study.

Lines115-116: “We observed that when the dehydrogenase, encoded by FadE, is strongly expressed, it elevates the rate of cellular H2O2 production.” Is there any information or did you try the rate of cellular H2O2 production in absence of FadE?

Line130-131: What is the source of dodecanoic acid? What concentration of ethanol was used to prepare dodecanoic acid/ and at what ratio to the LB medium ethanol was used (what would be final ethanol concentration in LB growth medium)? For H2O2 production studies in various genetic backgrounds, why you did not use WT cells along with media as controls?

Table1: write the full form of NEB. NO need to write “REF” just numbering is enough. Check for accurateness of the “Reference” column, which should have appropriate references/sources only.

Figure 6 legend, provide full forms of DTT and DHLA.

Reviewer's Comments 2:

MS # PONE-D-24-26458 intends to identify major site of ROS generation, which is ubiquitously loosely attributed to FADH2 oxidation during aerobic respiration. However, there are many oxidative pathways wherein FADH2 generates and enters the oxidative electron transport system for ATP production and electron leakage responsible for ROS generation. With the help of E. coli fadE (Acyl-CoA dehydrogenase) mutant, the generation of hydrogen peroxide has been monitored during the in vitro βoxidation of dodecanoic acid, a fatty acid, for the mutant was incapable of scavenging the generated hydrogen peroxide. The authors are inclined to believe that βoxidation generates the major source of ROS within the limits of their experimental setup. Of course, the findings are significant and prompts the further advanced work.

Experimental design and data generation is sound and appears to be reproducible. Results and Discussion are posited logically. However, the MS needs through scrutiny for standard usage of English language e.g., use “must arise” in place of “must be arise” in Line 45-46 “However, the investigators quickly inferred that these oxygen species must be arise inside cells and be capable of damaging the organism.”

Reviewers' comments:

Reviewer's Responses to Questions

**Comments to the Author**

1. Is the manuscript technically sound, and do the data support the conclusions?

Reviewer #1: Yes

Reviewer #2: Yes

2. Has the statistical analysis been performed appropriately and rigorously? 

Reviewer #1: Yes

Reviewer #2: Yes

3. Have the authors made all data underlying the findings in their manuscript fully available?

Reviewer #1: Yes

Reviewer #2: Yes

4. Is the manuscript presented in an intelligible fashion and written in standard English?

Reviewer #1: Yes

Reviewer #2: Yes

5. Review Comments to the Author

**Reviewer #1:** Review comment on “Evidence for endogenous hydrogen peroxide production by E. coli fatty acyl-CoA dehydrogenase”

General comment: This is a nice piece of work on detail understanding on how superoxides and H2O2 are produced continuously in bacterial cells. Results presented in figures but not described elaborately, indeed results are given partially as in typical “results & Discussion” section, while this article has again a separate discussion section. So, it is actually better combine both sections as “Results and Discussion”, and describe a little of the Figures as a result component of the section.

Specific comment:

Lines22-23: “To avoid this problem cells maintain high levels of scavenging enzymes” provide names of a few important scavenging enzymes relevant to this study.

Lines115-116: “We observed that when the dehydrogenase, encoded by FadE, is strongly expressed, it elevates the rate of cellular H2O2 production.” Is there any information or did you try the rate of cellular H2O2 production in absence of FadE?

Line130-131: What is the source of dodecanoic acid? What concentration of ethanol was used to prepare dodecanoic acid/ and at what ratio to the LB medium ethanol was used (what would be final ethanol concentration in LB growth medium)? For H2O2 production studies in various genetic backgrounds, why you did not use WT cells along with media as controls?

Table1: write the full form of NEB. NO need to write “REF” just numbering is enough. Check for accurateness of the “Reference” column, which should have appropriate references/sources only.

Figure 6 legend, provide full forms of DTT and DHLA.

**Reviewer #2: **MS # PONE-D-24-26458 intends to identify major site of ROS generation, which is ubiquitously loosely attributed to FADH2 oxidation during aerobic respiration. However, there are many oxidative pathways wherein FADH2 generates and enters the oxidative electron transport system for ATP production and electron leakage responsible for ROS generation. With the help of E. coli fadE (Acyl-CoA dehydrogenase) mutant, the generation of hydrogen peroxide has been monitored during the in vitro βoxidation of dodecanoic acid, a fatty acid, for the mutant was incapable of scavenging the generated hydrogen peroxide. The authors are inclined to believe that βoxidation generates the major source of ROS within the limits of their experimental setup. Of course, the findings are significant and prompts the further advanced work.

Experimental design and data generation is sound and appears to be reproducible. Results and Discussion are posited logically. However, the MS needs through scrutiny for standard usage of English language e.g., use “must arise” in place of “must be arise” in Line 45-46 “However, the investigators quickly inferred that these oxygen species must be arise inside cells and be capable of damaging the organism.”

6. PLOS authors have the option to publish the peer review history of their article (what does this mean?). If published, this will include your full peer review and any attached files.

Reviewer #1: No

Reviewer #2: **Yes: **Shamim A Ansari

---

## [Author Response · Author response to Decision Letter 0]

9 Aug 2024

Response to reviewers

Manuscript ID: PONE-D-24-26458

Reviewer's Comments 1:

Review comment on “Evidence for endogenous hydrogen peroxide production by E. coli fatty acyl-CoA dehydrogenase”

General comment: This is a nice piece of work on detail understanding on how superoxides and H2O2 are produced continuously in bacterial cells. Results presented in figures but not described elaborately, indeed results are given partially as in typical “results & Discussion” section, while this article has again a separate discussion section. So, it is actually better combine both sections as “Results and Discussion”, and describe a little of the Figures as a result component of the section.

Response: Thank you for your reading of our manuscript. As requested, we have combined the Results and Discussion sections.

Specific comment:

Lines22-23: “To avoid this problem cells maintain high levels of scavenging enzymes” provide names of a few important scavenging enzymes relevant to this study.

Response: the phase “scavenging enzymes” has been replaced with “superoxide dismutases, catalases, and peroxidases.”

Lines115-116: “We observed that when the dehydrogenase, encoded by FadE, is strongly expressed, it elevates the rate of cellular H2O2 production.” Is there any information or did you try the rate of cellular H2O2 production in absence of FadE?

Response: We did this experiment, as demonstrated in Figure 3. As expected, the absence of fadE has no impact on H2O2 formation in a fadR+ strain, because the enzyme is not induced under those conditions. Induction—and high H2O2 production—occurs in the fadR mutant. Accordingly, Figure 4 confirms that the growth defect created by the fadR mutation is reversed when fadE is deleted. 

Line130-131: What is the source of dodecanoic acid? What concentration of ethanol was used to prepare dodecanoic acid/ and at what ratio to the LB medium ethanol was used (what would be final ethanol concentration in LB growth medium)? For H2O2 production studies in various genetic backgrounds, why you did not use WT cells along with media as controls?

Response: The dodecanoic acid was purchased from Sigma and dissolved in absolute ethanol as a 50 mM stock solution before dilution into the working media. Thus, the final concentration of ethanol was 0.1 %, which is not toxic to the bacterium. This information has now been added to the Materials & Methods. 

Wild-type cells (MG1655) do not release any H2O2, and we reproduced this result. This observation is now stated in the section of the Materials & Methods section that presents the rationale for doing these experiments in an HPX- (scavenging-deficient) background. 

Table1: write the full form of NEB. NO need to write “REF” just numbering is enough. Check for accurateness of the “Reference” column, which should have appropriate references/sources only.

Response: Done. 

Figure 6 legend, provide full forms of DTT and DHLA.

Response: Done.

Reviewer's Comments 2:

MS # PONE-D-24-26458 intends to identify major site of ROS generation, which is ubiquitously loosely attributed to FADH2 oxidation during aerobic respiration. However, there are many oxidative pathways wherein FADH2 generates and enters the oxidative electron transport system for ATP production and electron leakage responsible for ROS generation. With the help of E. coli fadE (Acyl-CoA dehydrogenase) mutant, the generation of hydrogen peroxide has been monitored during the in vitro βoxidation of dodecanoic acid, a fatty acid, for the mutant was incapable of scavenging the generated hydrogen peroxide. The authors are inclined to believe that βoxidation generates the major source of ROS within the limits of their experimental setup. Of course, the findings are significant and prompts the further advanced work.

Experimental design and data generation is sound and appears to be reproducible. Results and Discussion are posited logically. 

However, the MS needs through scrutiny for standard usage of English language e.g., use “must arise” in place of “must be arise” in Line 45-46 “However, the investigators quickly inferred that these oxygen species must be arise inside cells and be capable of damaging the organism.”

Response: Done.

---

## [Decision Letter · Decision Letter 1]

22 Aug 2024

Evidence for endogenous hydrogen peroxide production by  E. coli  fatty acyl-CoA dehydrogenase

PONE-D-24-26458R1

Dear Dr. Sirithanakorn,

We’re pleased to inform you that your manuscript has been judged scientifically suitable for publication and will be formally accepted for publication once it meets all outstanding technical requirements.

Kind regards,

Vipin Chandra Kalia, FNASc, FAMI

Academic Editor

PLOS ONE

Additional Editor Comments (optional):

There is a minor suggestion which may be rectified at the Galley Proof stage.

Specific comment:

The following comment was answered in the response letter but the information included in the materials and methods section was incomplete. Please re-write providing the detail rationale of not using WT cells. It is also not true that any living organism with Respiratory Electron Transport (RET) system wouldn’t generate H2O2, it may be undetectable levels.

“For H2O2 production studies in various genetic backgrounds, why you did not use WT cells along with media as controls?”

Reviewers' comments:

Reviewer's Responses to Questions

**Comments to the Author**

1. If the authors have adequately addressed your comments raised in a previous round of review and you feel that this manuscript is now acceptable for publication, you may indicate that here to bypass the “Comments to the Author” section, enter your conflict of interest statement in the “Confidential to Editor” section, and submit your "Accept" recommendation.

Reviewer #1: All comments have been addressed

Reviewer #2: All comments have been addressed

2. Is the manuscript technically sound, and do the data support the conclusions?

Reviewer #1: Yes

Reviewer #2: Yes

3. Has the statistical analysis been performed appropriately and rigorously? 

Reviewer #1: N/A

Reviewer #2: Yes

4. Have the authors made all data underlying the findings in their manuscript fully available?

Reviewer #1: Yes

Reviewer #2: Yes

5. Is the manuscript presented in an intelligible fashion and written in standard English?

Reviewer #1: Yes

Reviewer #2: Yes

6. Review Comments to the Author

Reviewer #1: Specific comment:

The following comment was answered in the response letter but the information included in the materials and methods section was incomplete. Please re-write providing the detail rationale of not using WT cells. It is also not true that any living organism with Respiratory Electron Transport (RET) system wouldn’t generate H2O2, it may be undetectable levels.

“For H2O2 production studies in various genetic backgrounds, why you did not use WT cells along with media as controls?”

Reviewer #2: The revised text incorporates suggestions/queries raised by both reviewers and has improved significantly. I trust the revised MS deserves to be accepted.

7. PLOS authors have the option to publish the peer review history of their article (what does this mean?). If published, this will include your full peer review and any attached files.

Reviewer #1: No

Reviewer #2: **Yes: **Shamim A Ansari

---

## [Editor Report · Acceptance letter]

26 Aug 2024

PONE-D-24-26458R1 

PLOS ONE

Dear Dr. Sirithanakorn, 

I'm pleased to inform you that your manuscript has been deemed suitable for publication in PLOS ONE. Congratulations! Your manuscript is now being handed over to our production team.

Kind regards, 

on behalf of

Dr. Vipin Chandra Kalia 

Academic Editor

PLOS ONE